# Diagnostic efficacy of the magnetic resonance T1w/T2w ratio for the middle cerebellar peduncle in multiple system atrophy and spinocerebellar ataxia: A preliminary study

Jiaqi Wang[1], Atsuhiko Sugiyama[1]*, Hajime Yokota[2], Shigeki Hirano[1], Graham Cooper[3,4,5,6,7], Hiroki Mukai[8], Kenji Ohira[8], Kyosuke Koide[1], Shoichi Ito[1,9], Carsten Finke[5,7,10], Alexander U. Brandt[4,11], Friedemann Paul[3,4,5], Satoshi Kuwabara[1]

1 Department of Neurology, Graduate School of Medicine, Chiba University, Chiba, Japan, 2 Department of Diagnostic Radiology and Radiation Oncology, Graduate School of Medicine, Chiba University, Chiba, Japan, 3 Experimental and Clinical Research Center, Max Delbrueck Center for Molecular Medicine and Charité-Universitätsmedizin Berlin, Corporate Member of Freie Universität Berlin, Humboldt-Universität zu Berlin and Berlin Institute of Health, Berlin, Germany, 4 NeuroCure Clinical Research Center, Charité-Universitätsmedizin Berlin, Berlin, Germany, 5 Einstein Center for Neurosciences, Berlin, Germany, 6 Department of Experimental Neurology and Center for Stroke Research, Berlin, Charité–Universitätsmedizin Berlin, Berlin, Germany, 7 Department of Neurology, Charité-Universitätsmedizin Berlin, Berlin, Germany, 8 Department of Radiology, Chiba University Hospital, Chiba, Japan, 9 Department of Medical Education, Graduate School of Medicine, Chiba University, Chiba, Japan, 10 Berlin School of Mind and Brain, Humboldt-Universität zu Berlin, Berlin, Germany, 11 Department of Neurology, University of California, Irvine, California, United States of America

* asugiyama@chiba-u.jp

**Data Availability Statement:** All relevant data are within the paper and its Supporting information files.

## Abstract

### Background

The standardized T1-weighted/T2-weighted (sT1w/T2w) ratio for the middle cerebellar peduncle (MCP) has been reported to be sensitive for detecting degenerative changes in the cerebellar subtype of multiple system atrophy (MSA-C), even in the early stages. We aimed to investigate the diagnostic value of the MCP sT1w/T2w ratio for differentiating between MSA-C and spinocerebellar ataxia (SCA).

### Methods

We included 32 MSA-C, 8 SCA type 3 (SCA3), 16 SCA type 6 (SCA6) patients, and 17 controls, and the MCP sT1w/T2w ratio was analyzed using a region-of-interest approach. The diagnostic performance of the MCP sT1w/T2w ratio in discriminating among MSA-C, SCA3, and SCA6 was assessed and compared with diagnosis based on visual interpretation of MCP hyperintensities and the "hot cross bun" (HCB) sign.

### Results

MCP sT1w/T2w ratio values were markedly lower in patients with MSA-C than in those with SCA3, those with SCA6, and controls ($p < 0.001$). The MCP sT1w/T2w ratio showed high diagnostic accuracy for distinguishing MSA-C from SCA3 (area under curve = 0.934), SCA6

**Funding:** This work was partly supported by Grants-in-Aid from the Research Committee of Ataxia, Health Labor Sciences Research Grant, the Ministry of Health, Labor and Welfare, Japan (JPMH20FC1041). The funders had no role in study design, data collection and analysis, decision to publish, or preparation of the manuscript.

**Competing interests:** The authors have declared that no competing interests exist.

**Abbreviations:** ANCOVA, One-way analysis of covariance; ANOVA, Univariate 1-way analysis of variance; AUC, Area under the curve; GCI, Glial cytoplasmic inclusion; HCB, Hot cross bun; ICC, Intraclass correlation coefficient; MCP, Middle cerebellar peduncle; MRI, Magnetic resonance imaging; MSA, Multiple system atrophy; ROC, Receiver operating characteristic; SCA, Spinocerebellar ataxia; sT1w/T2w ratio, Standardized T1w/T2w ratio; T1w/T2w ratio, The ratio of the signal intensity of T1-weighted and T2-weighted images.

(area under curve = 0.965), and controls (area under curve = 0.980). The diagnostic accuracy of the MCP sT1w/T2w ratio for differentiating MSA-C from SCA3 or SCA6 (90.0% for MSA-C vs. SCA3, and 91.7% for MSA-C vs. SCA6) was comparable to or superior than that of visual interpretation of MCP hyperintensities (80.0–87.5% in MSA-C vs. SCA3 and 87.6–97.9% in MSA-C vs. SCA6) or the HCB sign (72.5–80.0% in MSA-C vs. SCA3 and 77.1–93.8% in MSA-C vs. SCA6).

## Conclusions

The MCP sT1w/T2w ratio might be a sensitive imaging-based marker for detecting MSA-C-related changes and differentiating MSA-C from SCA3 or SCA6.

## Introduction

Multiple system atrophy (MSA) is an adult-onset, sporadic degenerative disease of the nervous system that is characterized by a combination of autonomic nervous dysfunction, parkinsonism, cerebellar ataxia, and pyramidal signs. MSA is principally divided into two clinical subtypes: MSA with parkinsonism as the predominant manifestation (MSA-P) and MSA with cerebellar ataxia as the predominant manifestation (MSA-C). According to the second consensus statement on the diagnosis of MSA, diagnostic criteria for MSA-C span three categories: definite MSA-C, probable MSA-C, and possible MSA-C [1]. Pons or middle cerebellar peduncle (MCP) atrophy seen on brain magnetic resonance imaging (MRI) is defined as one of the additional features required for a diagnosis of possible MSA-C [1]. Other MRI features, such as the "hot cross bun" (HCB) sign and hyperintensities in the MCP on T2-weighted images (MCP hyperintensities), have also been described in MSA-C patients [2], and it has been proposed that these MRI features can be added to the list of additional features of possible MSA-C depending on their diagnostic utility [3]. MCP hyperintensities are thought to reflect myelin loss in the MCP [4] and have been repeatedly reported to be useful in the diagnosis of MSA-C [4–10]; however, their evaluation is qualitative, is subject to interpreter bias, and have limited sensitivity.

Recent studies have demonstrated the utility and unique quantitative contrast provided by the ratio of the signal intensity of T1-weighted and T2-weighted images (T1w/T2w ratio) as they possess high test-retest reliability and are sensitive to neurodegenerative changes [11–13]. Recently, Misaki et al., have proposed standardization of the T1w/T2w ratio (sT1w/T2w ratio), as it would allow meaningful comparison between subjects and scanners by not only creating scaled intensity values but also correcting for inhomogeneities in receiver coil sensitivity [14]. Evidence on the sensitivity of the sT1w/T2w ratio for detecting disease related changes, for example, in multiple sclerosis and MSA, continues to increase, [15–17], and we have previously reported that using the sT1w/T2w ratio for evaluating the MCP in MSA-C patients can help detect early MSA-C related degenerative changes as it shows extremely high diagnostic accuracy in distinguishing between MSA-C and healthy individuals [16]. However, it is not known if the MCP sT1w/T2w ratio is useful for differentiating between MSA-C and spinocerebellar ataxia (SCA), as cerebellar ataxia is a predominant motor symptom in both conditions. SCA types 3 (SCA3) and 6 (SCA6) are common subtypes of autosomal dominant cerebellar ataxia [18], and, in Japan, SCA3 is the most common subtype in which MCP hyperintensities and the HCB sign can be observed [19].

Therefore, to assess the diagnostic utility of the MCP sT1w/T2w ratio in detecting MSA-C and SCA, we calculated the MCP sT1w/T2w ratio for MSA-C, SCA type 3 (SCA3), and SCA type 6 (SCA6), and compared the diagnostic accuracy of MCP sT1w/T2w ratio with that of visual interpretation of MCP hyperintensities and the HCB sign.

## Materials and methods

### Subjects

This retrospective study was approved by the Institutional Review Board of the Chiba University Graduate School of Medicine and the need for informed consent was waived. The inclusion criteria for patients with MSA-C were as follows: 1.5-T MR images acquired between April 2012 and March 2020, and a diagnosis of clinically confirmed, probable MSA-C based on criteria listed in the second consensus statement [1]. The inclusion criteria for patients with SCA3 and SCA6 were genetically confirmed SCA3 or SCA6 diagnosis and 1.5-T MR images acquired between April 2012 and March 2020. SCA3 and SCA6 patients with a disease duration of 10 years or more were excluded because differentiation from MSA-C is not needed such patients due to reduced life expectancy in MSA. The inclusion criteria for controls were as follows: patients who were referred to our hospital with complaints of headache, dizziness, or lightheadedness; those who had no neurological abnormalities; and those who underwent 1.5-T MRI between April 2012 and March 2020. The study cohort included 32 patients with MSA-C, 8 patients with SCA3, 16 patients with SCA6, and 17 controls. None of the patients had any other central nervous system disorders. Medical records of all patients were reviewed and data on age at MRI, age at onset, and disease duration from onset to MRI were retrieved.

### MRI acquisition

All MR images were acquired using a 1.5-T MRI system (Genesis Signa; GE Healthcare) with a 12-channel head coil. The MRI parameters for T1WI were as follows: 3D-IR-SPGR; sagittal plane; TR, 8 ms; TE, 3 ms; flip angle, 15˚; FOV, 220 × 220 mm; matrix, 256 × 256; voxel size, 0.86 × 0.86 × 1.5 mm; number of slices, 248; and number of averages, 1. Meanwhile, those for T2WI were as follows: 2D-TSE; TR, 4000 ms; TE, 106 ms; FOV, 220 × 220 mm; matrix, 320 × 256; voxel size, 0.43 × 0.43 × 6 mm; interslice gap, 1.5 mm; number of slices 24; and number of averages 1.

### MRI preprocessing

All T1w and T2w images were preprocessed as previously described before the sT1w/T2w ratio was calculated [16]; specifically, the intensity inhomogeneity correction was applied to the T1w and T2w images using *N4BiasFieldCorrection* [20]. 3D T1w images were linearly co-registered with 2D axial T2w images using SPM12. Brain masks were created by skull-stripping the co-registered T1w images using the Brain Extraction Tool with FSL (version 5.0.11) [21] and binarizing it with FSLmaths. White matter and gray matter brain masks were generated using FMRIB Automatic Segmentation Tool (FSL FAST) on the co-registered T1w image [22].

### T1w/T2w ratio and standardized T1w/T2w ratio

Median intensity values of T1w and T2w images, in both white and gray matter masks from each subject, were calculated using FSLstats. To obtain sT1w/T2w ratio, first, a scaling factor was calculated by dividing median gray matter intensity value in T1w images by the median gray matter intensity value in the T2w image. A scaled T2w image (sT2) was then created by multiplying the T2w image by the scaling factor. Finally, the sT1w/T2w ratio was calculated

using the following equation developed by Misaki et al [14].

$$s\frac{T1w}{T2w}ratio = \frac{T1w - sT2}{T1w + sT2}$$

Schematic representation of the pipeline for creating a sT1w/T2w ratio map is described in Fig 1. The sT1w/T2w ratio map for each subject was registered in the Montreal Neurological Institute 152 space [23] using Advanced Normalization Tools (ANTs) [24]. Each image was spatially smoothed with an 8-mm full-width at half-maximum Gaussian Kernel. Regions of interest in the MCP were defined bilaterally on normalized sT1w/T2w ratio maps using a validated probabilistic 3D atlas of the cerebellar white matter structure [25] and SPM in Matlab 2014a, as described previously [14]. Parcellation at a 90% probability threshold was used. Atlas registration accuracy was visually verified using the registration tool of SPM12 in Matlab 2014a. The mean value of median sT1w/T2w ratio values in the left and right MCP regions was used as the MCP sT1w/T2w ratio value for each subject.

## Visual interpretation of MCP hyperintensities and the HCB sign

Two board-certified neuroradiologists (H.M. and K.O.) and a board-certified neurologist (A. S.), who were blinded to the clinical data, independently evaluated MCP hyperintensities and the HCB sign for each subject, except for controls. MCP hyperintensities were marked as present or absent, where present was defined as "when high intensities relative to that of the brainstem and cerebellum in the adjacent white matter were observed on at least one side of the MCP" (Fig 2). The HCB sign was graded as described in a previous study [26], where 0 denoted no changes; 1 indicated initial appearance or presence of a clear vertical T2 high-intensity; and 2 indicated initial appearance of a horizontal line along with a vertical line or the presence of clear horizontal and vertical lines in the ventral pons (Fig 2). Here both grade 1 and grade 2 were defined as positive for HCB sign.

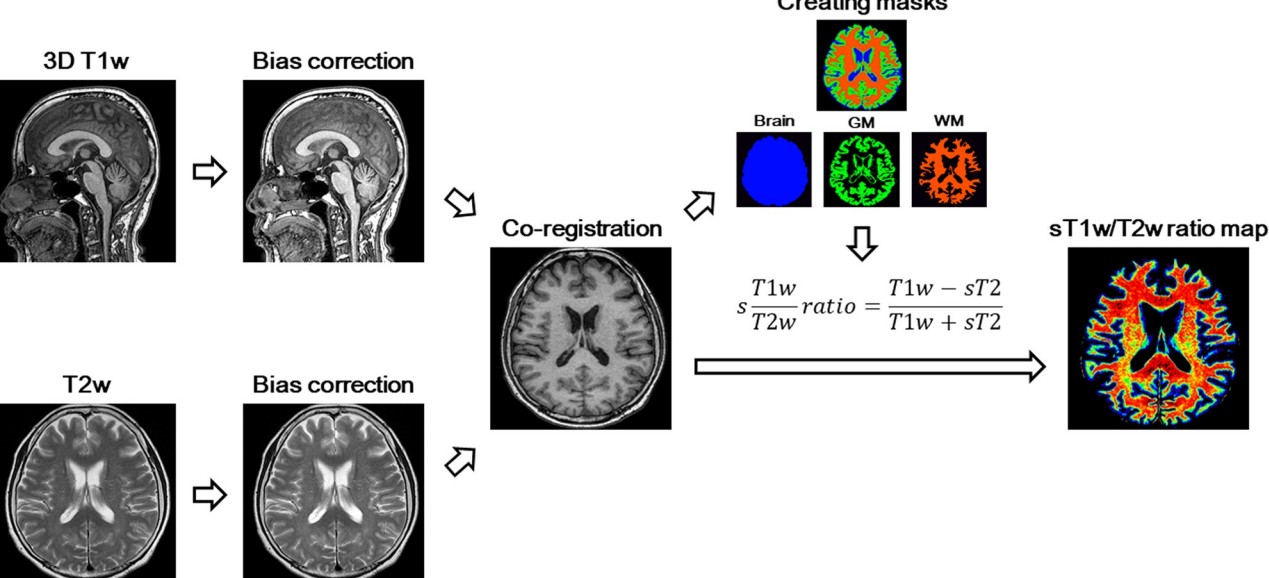

**Fig 1. Schematic representation of the pipeline for creating a sT1w/T2w ratio map.** T1w, T1-weighted; T2w, T2-weighted; GM, gray matter: WM, white matter; sT2, a scaled T2-weighted image.

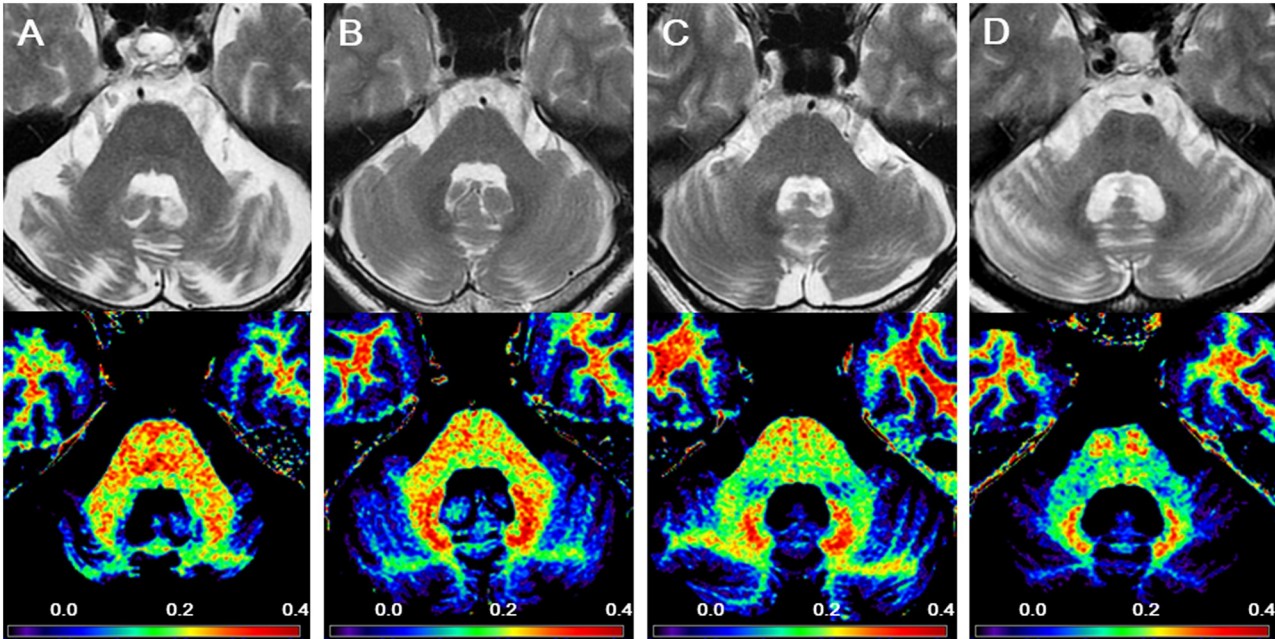

**Fig 2. Typical Middle Cerebellar Peduncle (MCP) hyperintensities and Hot Cross Bun (HCB) sign on magnetic resonance imaging in cerebellar-type multiple system atrophy (MSA-C) and spinocerebellar ataxia type 3 (SCA3) and 6 (SCA6).** (A) Grade 0 (negative) HCB sign and absent MCP hyperintensities in a 74-year-old man with a SCA6 disease duration of 2.3 years. (B) Grade 1 HCB sign and absent MCP hyperintensities in a 66-year-old man with SCA3 and a disease duration of 3.8 years. (C) Grade 1 HCB sign and MCP hyperintensities in a 57-year-old man with MSA-C and a disease duration of 0.9 years. (D) Grade 2 HCB sign and MCP hyperintensities in a 62-year-old woman with MSA-C and a disease duration of 2.8 years. Upper row: axial T2-weighted images. Lower row: standardized T1-weighted/T2-weighted ratio maps.

A board-certified neuroradiologist (H. Y.), who was blinded to the clinical data, semiquantitatively evaluated the extent of MCP hyperintensities in the patients with MSA by applying the following scores: 0 = absent, 1 = mild, 2 = moderate, and 3 = severe.

## MCP volume analysis

The MCP volume was measured in all subjects by exploiting a validated probabilistic 3D atlas of the cerebellar white matter structure [26]. Parcellation at a 90% probability threshold was used. The atlas of the MCP was transformed from the Montreal Neurological Institute 152 space [23] to the individual subject's space using the deformation parameters obtained from the normalization procedure using ANTs [24]. The mean volume value in the left and right MCPs was used as the MCP volume for each subject. For further analyses, the MCP volume was normalized by each subject's intracranial volume. Values represent the ratio of the volume of the MCP divided by the subject-specific total intracranial volume (written as $m \times 10^{-3}$).

## Statistical analyses

All statistical analyses, except for receiver operating characteristic (ROC) curve analyses, were performed using Statistical Package for the Social Sciences (version 25.0; SPSS Inc., IBM Corp., Chicago, IL, USA). The ROC curve analyses and Smirnov–Grubb's test were performed using EZR (Saitama Medical Center, Jichi Medical University, Saitama, Japan), which is a graphical user interface for R (The R Foundation for Statistical Computing, Vienna, Austria, version 2.13.0) [27]. More precisely, EZR is a modified version of R commander (version 1.6–3) designed to add statistical functions frequently used in biostatistics. Demographic data of

the patients with MSA-C, SCA3, and SCA6 and controls were compared using the chi-square ($\chi^2$) test for sex and univariate one-way analysis of variance for age at MRI and onset. Differences in disease duration among the three groups of patients were analyzed using the Kruskal–Wallis test, followed by post-hoc Mann–Whitney $U$ tests adjusted for multiple comparisons (Bonferroni correction). For evaluating differences in MCP sT1w/T2w ratio values and MCP volume among the four groups, a one-way analysis of covariance was used using age as the covariate. The ability of the MCP sT1w/T2w ratio values and MCP volume to differentiate between groups of patients was assessed using ROC curve analyses, and the optimal cutoff point was determined as the point on the ROC curve closest to the upper left corner, as used by Holmes [28]. The DeLong test was used to compare the area under curves (AUCs) between the MCP sT1w/T2w ratio and MCP volume to differentiate among the four groups. Diagnostic accuracy of the MCP sT1w/T2w ratio and visual interpretation of MCP hyperintensities and HCB sign provided by three reviewers were compared using binomial tests. The intraclass correlation coefficient (ICC) was calculated to assess inter-reviewer variability in visual interpretation of the MCP hyperintensities and the HCB sign. Spearman correlation analysis was performed to evaluate the relationship between the extent of MCP hyperintensities assessed semiquantitatively and the MCP sT1w/T2w ratio values in patients with MSA-C. Pearson correlation analysis was performed to evaluate the relationship between the MCP sT1w/T2w ratio values and MCP volume in patients with MSA-C. The Smirnov–Grubb's test was used for evaluating the outliers of the MCP sT1w/T2w ratio value in each subject group. $P < 0.05$ was considered statistically significant.

## Results

Demographic and clinical data of patients with MSA-C, SCA3, and SCA6 are summarized in Table 1.

**Table 1. Demographic and clinical data of the patients with cerebellar subtype multiple system atrophy (MSA-C), spinocerebellar ataxia type 3 (SCA3), and type 6 (SCA6) and controls.**

| Group (n) | Sex Distribution[a] | Age at MRI[b] | Disease Duration[c] | Age at Onset[b] |
|---|---|---|---|---|
| | (Male/Female) | (years, mean ± SD) | (years, median, range) | (years, mean ± SD) |
| MSA-C (n = 32) | 19/13 | 62.8 ± 9.4 | 2.3 (0.8–5.9) | 60.4 ± 9.4 |
| SCA3 (n = 8) | 2/6 | 51.4 ± 16.7 | 4.5 (0.8–5.7) | 47.3 ± 17.4 |
| SCA6 (n = 16) | 10/6 | 58.3 ± 13.7 | 3.5 (0.3–9.8) | 54.1 ± 12.3 |
| Control (n = 17) | 9/8 | 62.4 ± 10.8 | NA | NA |
| P-value for group comparisons | 0.350 | 0.078 | **0.026** | 0.069 |
| P-value for post-hoc group comparisons | | | | |
| MSA-C vs. SCA3 | NA | NA | 0.030 | NA |
| MSA-C vs. SCA6 | NA | NA | 0.034 | NA |
| MSA-C vs. control | NA | NA | NA | NA |
| SCA3 vs. SCA6 | NA | NA | 0.742 | NA |
| SCA3 vs. control | NA | NA | NA | NA |
| SCA6 vs. control | NA | NA | NA | NA |

MRI, magnetic resonance imaging; NA, not applicable.

[a] Chi-squared test.

[b] Parametric tests (univariate 1-way analysis of variance [ANOVA]).

[c] Nonparametric tests (Kruskal-Wallis 1-way ANOVA with post-hoc Mann-Whitney $U$ tests adjusted for multipe comparisons; $P$ value for comparisons of Disease Duration: $P < 0.05/3 = 0.0167$).

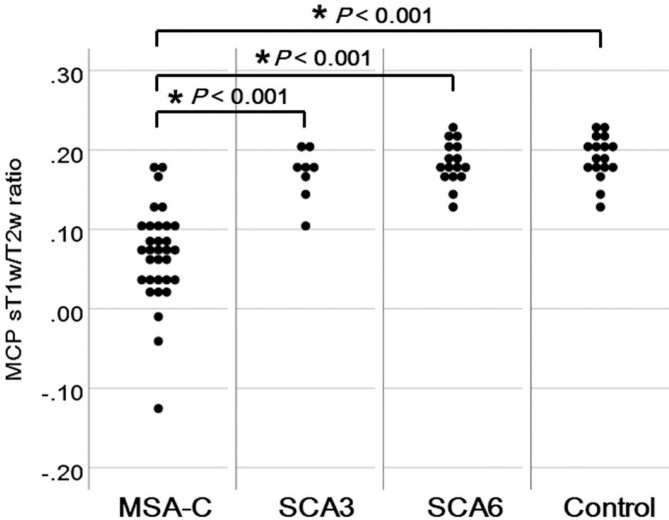

**Fig 3. Comparison of the Middle Cerebellar Peduncle (MCP) standardized T1-weighted/T2-weighted (sT1w/T2w) ratio among patients with cerebellar subtype multiple system atrophy (MSA-C), spinocerebellar ataxia type 3 (SCA-3), and spinocerebellar ataxia type 6 (SCA-6) and controls.** The MCP sT1w/T2w ratio values were significantly lower in patients with MSA-C than those in patients with SCA-3 and SCA-6 and controls ($^*p < 0.001$).

The MCP sT1w/T2w ratio values were significantly lower in patients with MSA-C than those in patients with SCA3 and SCA6 and controls ($0.07 \pm 0.06$ vs. $0.17 \pm 0.03$, $0.18 \pm 0.03$, and $0.19 \pm 0.03$, respectively; $p < 0.001$) (Fig 3). The AUC for differentiating MSA-C from SCA-3 was 0.934 (95% confidence interval [CI], 0.854–1.000), with a sensitivity of 0.906 and specificity of 0.875. The AUC value for differentiating between MSA-C and SCA-6 was 0.965 (95% CI, 0.922–1.000), with a sensitivity of 0.906 and a specificity of 0.938. The AUC for differentiating MSA-C from controls was 0.980 (95% CI, 0.951–1.000), with a sensitivity of 0.906 and specificity of 0.941. The AUC for differentiating SCA-3 from SCA-6 was 0.633 (95% CI, 0.388–0.878), with a sensitivity of 0.750 and specificity of 0.625 (Fig 4). Even after excluding one patient with MSA-C with an outlier MCP sT1w/T2w ratio value, the MCP sT1w/T2w ratio values in patients with MSA-C were significantly lower than those in patients with SCA3 and SCA6 and controls ($0.07 \pm 0.05$ vs. $0.17 \pm 0.03$, $0.18 \pm 0.03$, and $0.19 \pm 0.03$, respectively; $p < 0.001$).

Sensitivity, specificity, positive and negative predictive values, and overall correct classification for each reviewer for visual evaluation of MCP hyperintensities and the HCB sign are shown in Table 2. MCP hyperintensities were not found in SCA6 patients and the sensitivity of MCP hyperintensities in MSA-C patients ranged from 84.4%–96.9%. Grade 2 HCB sign was highly specific for differentiating MSA-C from SCA3 and SCA6 but with moderate sensitivity. Grade 1 or 2 HCB sign (vertical or cruciform hyper-intensity) was highly sensitive (87.5%–100%) for MSA-C. However, even though grade 1 or 2 HCB sign could be observed in SCA3 patients, it was characterized by relatively low specificity for differentiating between MSA-C and SCA3. ICC values demonstrated substantial inter-reviewer agreement for visual interpretation of the MCP hyperintensities and the HCB sign (0.762 and 0.729, respectively; all $p < 0.001$). In patients with MSA-C, a significant inverse correlation between the extent of MCP hyperintensities and MCP sT1w/T2w ratio was observed ($r = -0.832$; $p < 0.001$).

Diagnostic accuracy of the MCP sT1w/T2w ratio, visual interpretation of the MCP hyperintensities, and the HCB sign are listed in Table 3, and we found that the diagnostic performance

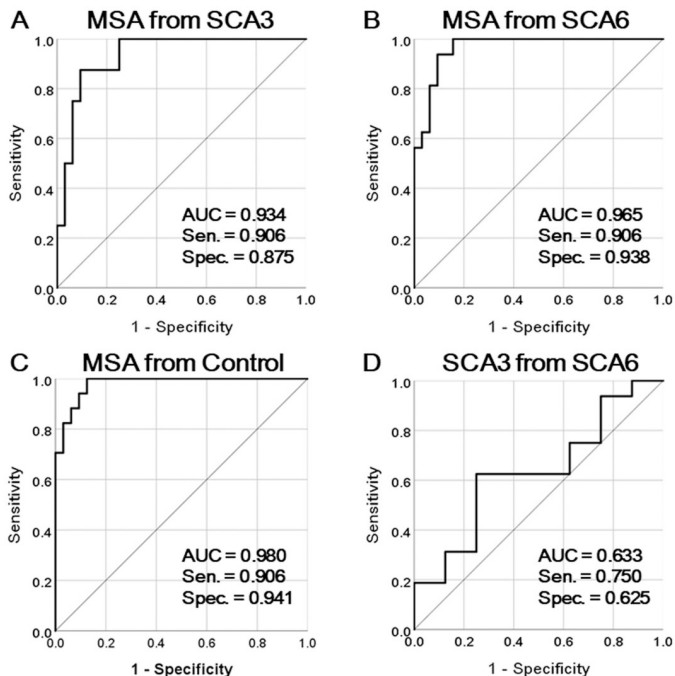

**Fig 4. Receiver Operating (ROC) curves for the MCP sT1w/T2w ratio showing area under the curve (AUC), sensitivity (Sen), and specificity (Spec).** A, B, and C are the ROC curves distinguishing cerebellar subtype multiple system atrophy (MSA-C) from spinocerebellar ataxia type 3 (SCA3) (A), spinocerebellar ataxia type 6 (SCA-6) (B), and controls (C). D is the ROC curve distinguishing SCA-3 from SCA-6.

of the MCP sT1w/T2w ratio for differentiating MSA-C from SCA3, SCA6, and controls was comparable to or better than that of visual interpretation of MCP hyperintensities and the HCB sign for all three reviewers. However, the diagnostic performance of visual interpretation of MCP hyperintensities and the HCB sign for differentiating SCA3 from SCA6 was comparable to or better than that of the MCP sT1w/T2w ratio.

The MCP volume, normalized by each subject's intracranial volume, was significantly lower in patients with MSA-C patients than those in patients with SCA6 and controls (0.20 ± 0.06 vs. 0.31 ± 0.05 and 0.34 ± 0.05, respectively; $p < 0.001$). Moreover, the MCP volume was significantly lower in patients with SCA3 than those in patients with SCA6 and controls (0.23 ± 0.04 vs. 0.31 ± 0.05 and 0.34 ± 0.05, respectively; $p < 0.001$; $p = 0.01$ and $p < 0.001$).

The AUC value for differentiating MSA-C from SCA3 was 0.664 (95% CI, 0.486–0.842), with a sensitivity of 0.750 and specificity of 0.625. The AUC for differentiating MSA-C from SCA6 was 0.924 (95% CI, 0.852–0.995), with a sensitivity of 0.844 and specificity of 0.875. The AUC for differentiating MSA-C from controls was 0.967 (95% CI, 0.925–1.000), with a sensitivity of 0.906 and specificity of 0.941). The AUC for differentiating SCA3 from SCA6 was 0.914 (95% CI, 0.802–1.000), with a sensitivity of 0.875 and specificity of 0.875. The AUC of the MCP sT1w/T2w ratio was greater than that of the MCP volume for differentiating MSA-C from SCA3 ($p < 0.001$). No significant differences were observed between the AUC of the MCP sT1w/T2w ratio and that of MCP volume for differentiating MSA-C from SCA6 and controls ($p = 0.234$ and 0.577, respectively). The AUC of the MCP volume was greater than that of the MCP sT1w/T2w ratio for differentiating SCA3 from SCA6 ($p = 0.020$). In patients with MSA-C, a significant correlation was observed between the MCP volume and MCP sT1W/T2W ratio ($r = 0.619$; $p < 0.001$).

**Table 2.** Visual evaluation of the Middle Cerebellar Peduncle (MCP) hyperintensities and hot cross bun sign for cerebellar subtype multiple system atrophy (MSA-C) and spinocerebellar ataxia type 3 (SCA3) and 6 (SCA6).

| | MSA vs. SCA3 | | | MSA vs. SCA6 | | | SCA3 vs. SCA6 | | |
|---|---|---|---|---|---|---|---|---|---|
| | Reviewer 1 | Reviewer 2 | Reviewer 3 | Reviewer 1 | Reviewer 2 | Reviewer 3 | Reviewer 1 | Reviewer 2 | Reviewer 3 |
| **MCP hyperintensities** | | | | | | | | | |
| Sensitivity, % (95% CI) | 90.6 | 96.9 | 84.4 | 90.6 | 96.9 | 84.4 | 25.0 | 87.5 | 12.5 |
| | (75.0–98.0) | (83.8–99.9) | (67.2–94.7) | (75.0–98.0) | (83.8–99.9) | (67.2–94.7) | (3.2–65.1) | (47.3–99.7) | (0.3–52.7) |
| Specificity, % (95% CI) | 75.0 | 12.5 | 87.5 | 100 | 100 | 100 | 100 | 100 | 100 |
| | (34.9–96.8) | (0.3–52.7) | (47.3–99.7) | (71.3–100) | (71.3–100) | (71.3–100) | (71.3–100) | (71.3–100) | (71.3–100) |
| Positive predictive value, % (95% CI) | 93.5 | 81.6 | 96.4 | 100 | 100 | 100 | 100 | 100 | 100 |
| | (78.6–99.2) | (65.7–92.3) | (81.7–99.9) | (82.8–100) | (83.8–100) | (81.7–100) | (9.4–100) | (47.3–100) | (1.3–100) |
| Negative predictive value, % (95% CI) | 66.7 | 50.0 | 58.3 | 84.2 | 94.1 | 76.2 | 72.7 | 94.1 | 69.6 |
| | (29.9–92.5) | (1.3–98.7) | (27.7–84.8) | (60.4–96.6) | (71.3–99.9) | (52.8–91.8) | (49.8–89.3) | (71.3–99.9) | (47.1–86.8) |
| Overall correct classification, % (95% CI) | 87.5 | 80.0 | 85.0 | 93.8 | 97.9 | 89.6 | 75.0 | 95.8 | 70.8 |
| | (73.2–95.8) | (64.4–90.9) | (70.2–94.3) | (82.8–98.7) | (88.9–99.9) | (77.3–96.5) | (53.3–90.2) | (78.9–99.9) | (48.9–87.4) |
| **Grade 2 HCB sign (cruciform hyper-intensity)** | | | | | | | | | |
| Sensitivity, % (95% CI) | 59.4 | 81.2 | 75 | 59.4 | 81.2 | 75 | 0 | 0 | 12.5 |
| | (40.6–76.3) | (63.6–92.8) | (56.6–88.5) | (40.6–76.3) | (63.6–92.8) | (56.6–88.5) | (0–48.2) | (0–48.2) | (0.3–52.7) |
| Specificity, % (95% CI) | 100 | 100 | 87.5 | 100 | 100 | 100 | 100 | 100 | 100 |
| | (51.8–100) | (51.8–100) | (47.3–99.7) | (71.3–100) | (71.3–100) | (71.3–100) | (71.3–100) | (71.3–100) | (71.3–100) |
| Positive predictive value, % (95% CI) | 100 | 100 | 96.0 | 100 | 100 | 100 | 0 | 0 | 100 |
| | (75.1–100) | (81.0–100) | (79.6–99.9) | (75.1–100) | (81.0–100) | (79.6–100) | (0–100) | (0–100) | (1.3–100) |
| Negative predictive value, % (95% CI) | 38.1 | 57.1 | 46.7 | 55.2 | 72.7 | 66.7 | 66.7 | 66.7 | 69.6 |
| | (18.1–61.6) | (28.9–82.3) | (21.3–73.4) | (35.7–73.6) | (49.8–89.3) | (44.7–84.4) | (44.7–84.4) | (44.7–84.4) | (47.1–86.8) |
| Overall correct classification, % (95% CI) | 67.5 | 85.0 | 77.5 | 72.9 | 87.5 | 83.3 | 66.7 | 66.7 | 70.8 |
| | (50.9–81.4) | (70.2–94.3) | (61.5–89.2) | (58.2–84.7) | (74.8–95.3) | (69.8–92.5) | (44.7–84.4) | (44.7–84.4) | (48.9–87.4) |
| **Grade 1 or 2 HCB sign (vertical or cruciform hyper-intensity)** | | | | | | | | | |
| Sensitivity, % (95% CI) | 87.5 | 93.8 | 100 | 87.5 | 93.8 | 100 | 87.5 | 100 | 100 |
| | (71.0–96.5) | (79.2–99.2) | (84.2–100) | (71.0–96.5) | (79.2–99.2) | (84.2–100) | (47.3–99.7) | (51.8–100) | (51.8–100) |
| Specificity, % (95% CI) | 12.5 | 0 | 0 | 93.8 | 93.8 | 31.2 | 93.8 | 93.8 | 31.2 |
| | (0.3–52.7) | (0–48.2) | (0–48.2) | (69.8–99.8) | (69.8–99.8) | (11.0–58.7) | (69.8–99.8) | (69.8–99.8) | (11.0–58.7) |
| Positive predictive value, % (95% CI) | 80.0 | 78.9 | 80.0 | 96.6 | 96.8 | 74.4 | 87.5 | 88.9 | 42.1 |
| | (63.1–91.6) | (62.7–90.4) | (64.4–90.9) | (82.2–99.9) | (83.3–99.9) | (58.8–86.5) | (47.3–99.7) | (51.8–99.7) | (20.3–66.5) |
| Negative predictive value, % (95% CI) | 20.0 | 0 | 0 | 78.9 | 88.2 | 100 | 93.8 | 100 | 100 |
| | (0.5–71.6) | (0–90.6) | (0–100) | (54.4–93.9) | (63.6–98.5) | (35.9–100) | (69.8–99.8) | (69.8–100) | (35.9–100) |
| Overall correct classification, % (95% CI) | 72.5 | 75.0 | 80.0 | 89.6 | 93.8 | 77.1 | 91.7 | 95.8 | 54.2 |
| | (56.1–85.4) | (58.8–87.3) | (64.4–90.9) | (77.3–96.5) | (82.8–98.7) | (62.7–88.0) | (73.0–99.0) | (78.9–99.9) | (32.8–74.4) |

**Table 3. Comparison of accuracy between the MCP sT1w/T2w ratio and visual evaluation by three reviewers.**

| | MSA vs. SCA3 | | MSA vs. SCA6 | | SCA3 vs. SCA6 | |
|---|---|---|---|---|---|---|
| | Accuracy | P value (compared with MCP sT1w/T2w ratio) | Accuracy | P value (compared with MCP sT1w/T2w ratio) | Accuracy | P value (compared with MCP sT1w/T2w ratio) |
| MCP sTw/T2w ratio | 90.0 (76.3–97.2) | – | 91.7 (80.0–97.7) | – | 66.7 (44.7–84.4) | – |
| MCP hyperintensities | | | | | | |
| Reviewer 1 | 87.5 | 0.371 | 93.8 | 0.428 | 75.0 | 0.125 |
| | (73.2–95.8) | | (82.8–98.7) | | (53.3–90.2) | |
| Reviewer 2 | 80.0 | **0.042** | 97.9 | 0.083 | 95.8 | **0.001** |
| | (64.4–90.9) | | (88.9–99.9) | | (78.9–99.9) | |
| Reviewer 3 | 85.0 | 0.206 | 89.6 | 0.368 | 70.8 | 0.161 |
| | (70.2–94.3) | | (77.3–96.5) | | (48.9–87.4) | |
| Grade 2 HCB sign | | | | | | |
| Reviewer 1 | 67.5 | < **0.001** | 72.9 | < **0.001** | 66.7 | 0.171 |
| | (50.9–81.4) | | (58.2–84.7) | | (44.7–84.4) | |
| Reviewer 2 | 85.0 | 0.206 | 87.5 | 0.205 | 66.7 | 0.171 |
| | (70.2–94.3) | | (74.8–95.3) | | (44.7–84.4) | |
| Reviewer 3 | 77.5 | **0.015** | 83.3 | **0.043** | 70.8 | 0.161 |
| | (61.5–89.2) | | (69.8–92.5) | | (48.9–87.4) | |
| Grade 1 or 2 HCB sign | | | | | | |
| Reviewer 1 | 72.5 | **0.001** | 89.6 | 0.368 | 91.7 | **0.004** |
| | (56.1–85.4) | | (77.3–96.5) | | (73.0–99.0) | |
| Reviewer 2 | 75.0 | **0.005** | 93.8 | 0.428 | 95.8 | **0.001** |
| | (58.8–87.3) | | (82.8–98.7) | | (78.9–99.9) | |
| Reviewer 3 | 80.0 | 0.042 | 77.1 | **0.002** | 54.2 | 0.072 |
| | (64.4–90.9) | | (62.7–88.0) | | (32.8–74.4) | |

## Discussion

We showed that the MCP sT1W/T2W ratio is highly sensitive and specific for distinguishing MSA-C from SCA3, SCA6, or controls. Additionally, the diagnostic accuracy of the MCP sT1w/T2w ratio for differentiating MSA-C from SCA3 or SCA6 was comparable or superior to that of visual interpretation of MCP hyperintensities or the HCB sign. Given the variability in visual interpretation of MCP hyperintensities and the HCB sign among the three reviewers seen in the current study, the MCP sT1w/T2w ratio appears to be a feasible, reliable, and clinically valuable imaging biomarker for differentiating MSA-C from SCA3 or SCA6.

Decreased MCP sT1W/T2W ratio values in patients with MSA-C might reflect demyelination and gliosis in the MCP due to MSA-C-related degeneration. Pathologically, the loss of axons, myelin, and reactive gliosis was observed in the MCP of patients with MSA-C [29]. The sT1W/T2W ratio is sensitive to myelin and decreases in demyelinating lesions [30, 31]. In contrast, inconsistent correlations with histology [32], myelin water imaging [12, 33], simultaneous tissue relaxometry, and magnetization transfer saturation index [34] suggest that the sT1W/T2W ratio is sensitive to not only myelin content but also other microstructural factors. Moreover, gliosis is presumed to affect the sT1W/T2W ratio in a downward direction, mainly by prolonged T1 relaxation time [35]. Therefore, although the sT1W/T2W ratio might not be specific for myelin content, it might be more sensitive than evaluating MCP hyperintensities on T2WI to detect degenerative processes in the MCP, such as demyelination and gliosis

associated with MSA-C. In this study, the MCP sT1W/T2W ratio had better diagnostic accuracy in discriminating MSA-C from SCA3 than MCP volume. This result is consistent with those of previous pathological studies, showing that gliosis and demyelination precede atrophy in MSA [36, 37].

The usefulness of the MCP sT1W/T2W ratio in discriminating MSA-C from SCA3 in this study might be due to differences in the degree of MCP degeneration in the two diseases. Glial cytoplasmic inclusions (GCIs) are found in oligodendroglial cells, are recognized neuropathological hallmarks of MSA, and their density significantly correlates with disease progression [38, 39]. One of the regions where GCI pathology progresses in the earliest phases of MSA-C is the pontocerebellar fibers in the MCP, and regional severity of GCI pathology has been shown to correlate with the severity of myelin loss [40]. In SCA3, pathological changes are mainly seen in the efferent dentatorubral system, which passes through the superior peduncle, and in the pallidosubthalamic system [41]. Further, although degenerative changes can also be observed in the MCP, they are less severe than those seen in the superior peduncle [42, 43]. Previous MRI studies have also reported that MCP hyperintensities, which reflect degenerative changes in the MCP, are less frequently seen in patients with SCA3 than in patients with MSA, and that they were rarely observed during the early stages of SCAs, including SCA3 [10, 44].

The substantial discriminatory power of the MCP sT1w/T2w ratio and visual interpretation of MCP hyperintensities and the HCB sign described here implies that they may also be useful for differentiating between SCA3 and SCA6. The pathological background for this may be the difference in MCP degeneration, which is observed to some extent in SCA3, as described above, but is rarely observed in SCA6. Previous pathoanatomical studies have identified the cerebellar cortex, the dentate nucleus, and the inferior olive as predilection sites in SCA6 [45, 46]. Although recent studies have shown that neurodegeneration in SCA6 is more widely distributed, it is less widespread and less severe than in SCA1, SCA2, SCA3, and SCA7 [47, 48]. Concurring with our results, a previous MRI study has reported that MCP hyperintensities were not observed in any of the patients with SCA6 [49]. The cohort of SCA3 and SCA6 patients included in this study is not large, and therefore, further studies that include data from a large cohort of SCA3 and SCA6 patients are needed to verify the usefulness of MCP sT1w/T2w ratio values and visual interpretation of MCP hyperintensities and the HCB sign in differentiating between SCA3 and SCA6.

This study has several limitations. The first is that 2D T2W images were co-registered with 3D T1W images, which do not have the same image resolution, and a 1.5-T MRI scanner with an inherently low signal-to-noise ratio was used; therefore, the results of the MCP sT1W/T2W ratio may be biased by noise. However, the MCP sT1W/T2W ratio values in patients with MSA-C correlated well with the extent of MCP hyperintensities and MCP volume, suggesting that the MCP sT1W/T2W ratio values measured in this study well reflected the degeneration of the MCP in MSA-C. Moreover, the diagnostic accuracy of the MCP sT1W/T2W ratio in discriminating MSA-C from controls in this study was comparable to that observed in a study using a 3-T MRI scanner [16]. Further studies employing 3D T1WI and T2WI sequences using a 3-T MRI scanner, which might allow better quantification of the sT1W/T2W ratio, are needed to confirm the results of this study. Second, we did not pathologically confirm the diagnosis of MSA, and thus, the possibility of a misdiagnosis in some cases cannot be excluded. Third, owing to the retrospective nature of this study, the relationship between the MCP sT1W/T2W ratio and rapid eye movement sleep behavior disorder or clinical scales for cerebellar ataxia, such as the International Cooperative Ataxia Rating Scale or Scale for the Assessment and Rating of Ataxia, could not be evaluated. Finally, the sample size of the SCA3 group was small (n = 8), and future prospective studies with larger sample sizes that evaluate clinical

scales for cerebellar ataxia are needed to validate the findings of this study and clarify the relationship between clinical severity of cerebellar ataxia and the MCP sT1W/T2W ratio.

## Conclusions

In conclusion, the MCP sT1W/T2W ratio might be a sensitive imaging-based marker for detecting MSA-C-related changes and useful for differentiating MSA-C from SCA3 or SCA6.

## Supporting information

**S1 Table. Details of the clinical characteristics of subjects.**
(XLSX)

## Author Contributions

**Conceptualization:** Atsuhiko Sugiyama, Graham Cooper, Satoshi Kuwabara.

**Data curation:** Atsuhiko Sugiyama, Hajime Yokota, Kyosuke Koide.

**Formal analysis:** Atsuhiko Sugiyama, Hiroki Mukai, Kenji Ohira.

**Methodology:** Atsuhiko Sugiyama, Graham Cooper.

**Supervision:** Atsuhiko Sugiyama.

**Validation:** Atsuhiko Sugiyama, Shoichi Ito, Carsten Finke, Alexander U. Brandt, Friedemann Paul.

**Visualization:** Atsuhiko Sugiyama, Hajime Yokota, Shigeki Hirano, Graham Cooper, Shoichi Ito, Carsten Finke, Alexander U. Brandt, Friedemann Paul.

**Writing – original draft:** Jiaqi Wang.

**Writing – review & editing:** Atsuhiko Sugiyama, Satoshi Kuwabara.

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
