## [Decision Letter · Decision Letter 0]

20 Sep 2021

PONE-D-21-24426Diagnostic efficacy of the magnetic resonance T1w/T2w ratio for the middle cerebellar peduncle in multiple system atrophy and spinocerebellar ataxiaPLOS ONE

Dear Dr. Sugiyama,

Thank you for submitting your manuscript to PLOS ONE. After careful consideration, we feel that it has merit but does not fully meet PLOS ONE’s publication criteria as it currently stands. Therefore, we invite you to submit a revised version of the manuscript that addresses the points raised during the review process. Major methodological concerns have been raised, especially by Reviewer 2. It is unlikely that these limitations/issues can be fully addressed with the data that has been presented. It would be highly desirable if you could include additional scans, though, that are acquired with a protocol that is more similar to what is typically done with T1w/T2w ratio to show that your results are comparable.

We look forward to receiving your revised manuscript.

Kind regards,

Niels Bergsland

Academic Editor

PLOS ONE

Journal Requirements:

2. We noticed you have some minor occurrence of overlapping text with the following previous publication, which needs to be addressed:

- https://link.springer.com/article/10.1007%2Fs00330-020-07521-1

The text that needs to be addressed involves the materials and methods section.

In your revision ensure you cite all your sources (including your own works), and quote or rephrase any duplicated text. Further consideration is dependent on these concerns being addressed.

Reviewers' comments:

Reviewer's Responses to Questions

**Comments to the Author**

1. Is the manuscript technically sound, and do the data support the conclusions?

Reviewer #1: Yes

Reviewer #2: Partly

2. Has the statistical analysis been performed appropriately and rigorously? 

Reviewer #1: Yes

Reviewer #2: No

3. Have the authors made all data underlying the findings in their manuscript fully available?

Reviewer #1: Yes

Reviewer #2: Yes

4. Is the manuscript presented in an intelligible fashion and written in standard English?

Reviewer #1: Yes

Reviewer #2: Yes

5. Review Comments to the Author

Reviewer #1: The authors wrote an outstanding manuscript entitled "Diagnostic efficacy of the magnetic resonance T1w/T2w ratio for the middle cerebellarpeduncle in multiple system atrophy and spinocerebellar ataxia". They concluded that the MCP sT1w/T2w ratio is a sensitive imaging-based marker for detecting MSA-C related changes and might be useful for differentiating MSA-C from SCA3 or SCA6. I agree with the authors and I have some questions and comments:

1- Was there a correlation between MRI findings suggestive of MSA-C and the presence of a REM sleep behavior disorder?

2-Was there a correlation between MRI findings suggestive of MSA-C and the SARA ataxia scale?

3-Without a doubt, a prospective study would be very suitable to analyze the results of this study in greater depth, as well as with a greater number of cases of MSA-C and the group of patients with SCAs 3 and 6.

Reviewer #2: This study sought to investigate the diagnostic value of standardized T1w/T2w ratio in the middle cerebellar peduncle (MCP) for differentiating between the cerebellar subtype of multiple system atrophy (MSA-C) and spinocerebellar ataxia (SCA). The authors conducted this study on 32 patients with MSA-C, 8 patients with SCA3and 16 patients with SCA6 at 1.5 T MRI. They claimed that sT1w/T2w in MCP is a highly sensitive imaging-based marker for detecting MSA-C related changes and might be useful for differentiating MSA-C from SCA3 or SCA6.  However, a couple of major and minor concerns need to be addressed as listed below: 

1.     One of the major concerns of the paper is the data used. The results might be biased mostly by noise. Data used in this work is 2D and image resolutions arenot the same for T1w and T2w images and data was collected with 1.5 T MRI whichinherently has a low signal-to-noise ratio.  

2.     The authors haven’t clearly mentioned what they are measuring with T1w/T2w and how sensitive the method is to tissue microstructure especially the myelin. T1w/T2w method is considered to be sensitive to myelin but depends on how T1w and T2w images are acquired as well as the image processing pipeline. For better quantification of T1w/T2w, 3D T1w and 3D T2w SPACE sequences are recommended. A recent work by Uddin et al (NMR in Biomedicine 2018) reported that TSE based T1w/T2w ratio is not a good measure to quantify myelin.

3.     The authors mentioned about T2 hyperintensities in MCP. I was wondering if they have any scores for WM hyperintensities or disease severity scores that can be correlated with T1w/T2w.

4.     MRI acquisition: data were acquired with a 1.5 T scanner. Please add information about the RF coil (i.e, 32-channel head coil), number of averages, number of slices.

5.     MRI preprocessing: “3D T1w images were linearly co-registered with 2D  axial T2w images using SPM12. Brain masks were created by skull-stripping the co-registered T1w images using the BrainExtraction Tool with FSL (version 5.0.11) [20] and binarizing it with FSLmaths.”I was wondering why did the authors use SPM for coregistration only. It's better to use FSL or SPM, sometime the header information is not properly copied from one platform to other. However, it's possible to copy the headers from spm to fsl properly.

6.     T1w/T2w calculation: The authors mentioned that for T1w/T2w quantification they used previously published technique by Misaki et al MRM 2014, but it is not clear how did they perform bias correction which is important for better quantification ofT1w/T2w. Did they use their script or just follow their approach (it looks there is no publicly available tool online by Misaki et al)? Depending on the bias correction technique, T1w/T2w maps may contain residual radiofrequency transmit field (B1+) biases, which maybe correlated with these variables of interest, leading to potentially spurious results.

7.     To measure diagnostic accuracy by Youden ROC method, I suspect about the accuracy of the measurement with this small sample size.

8.     Figure1: you haven’t mentioned what type of images (i.e., T2-weighted images) are in the figure. Please include the corresponding T1w/T2w maps with the signal intensity scale.

9.     Figure2: significant difference between MSA-C with SCA3 and SCA6 might be derived from the outliers of MSA-C data. I would use the outlier removal tool and compare the results.  

10.  Discussion section should be more organized.

11.  It would be great to show the comparison of volumetric measure of the MCP indifferent disease conditions.   

12.  Please add the small sample size as a limitation.

13.  “In conclusion, the MCP sT1w/T2w ratio is a highly sensitive imaging-based marker for detecting MSA-C related changes and might be useful for differentiating 274 MSA-C from SCA3 or SCA6.” With a relatively small sample size and poor quality of data (especially 2D T1w, T2w data, 1.5 mm vs. 6mm slice thickness collected at 1.5 T), it is tough to claim “highly sensitive imaging-based marker ….”

6. PLOS authors have the option to publish the peer review history of their article (what does this mean?). If published, this will include your full peer review and any attached files.

Reviewer #1: No

Reviewer #2: **Yes: **Md Nasir Uddin

---

## [Author Response · Author response to Decision Letter 0]

5 Nov 2021

04/11/2021

Prof. Emily Chenette

Editor-in-Chief

PLOS ONE

Dear Prof. Chenette:

We greatly appreciate your review of our manuscript and the helpful suggestions. We have now revised the manuscript according to the reviewers’ comments. Changes made in accordance with the reviewers’ comments are indicated by red text in the revised manuscript. In your email of September 20, you recommended including additional scans acquired with a protocol that is more similar to what is typically done with T1w/T2w ratio. However, scans with such protocol have not been taken in the patients with SCA3 or SCA6 in our department so far. Instead, we have added a control group to the current study and have shown that diagnostic accuracy in differentiating MSA from controls is comparable to a previous study using 3T MRI (Sugiyama A, et al. Eur Radiol 2021;31:4277-4284).

We hope that you will consider this revised version suitable for publication.

Sincerely,

Dr. Atsuhiko Sugiyama, MD, PhD

Department of Neurology, Graduate School of Medicine, Chiba University

Phone No: +81-43-226-2129

Fax No: +81-43-226-2160

Email Address: asugiyama@chiba-u.jp

 

Reply to Reviewer 1

We greatly appreciate your helpful comments and suggestions. Changes made in accordance with reviewers’ comments are indicated in red font in the revised manuscript.

#1 Was there a correlation between MRI findings suggestive of MSA-C and the presence of a REM sleep behavior disorder?

Reply: Unfortunately, there was no systematic evaluation of RBD using questionnaires or PSG in the patients with MSA included in this study. Moreover, some patients were not asked about their RBD since little attention had been paid to the relationship between RBD and MSA in the past. We have added a note to the Discussion section about this as a limitation, as follows.

Page 21, line 348-351

“Third, owing to the retrospective nature of this study, the relationship between the MCP sT1W/T2W ratio and rapid eye movement sleep behavior disorder or clinical scales for cerebellar ataxia, such as the International Cooperative Ataxia Rating Scale or Scale for the Assessment and Rating of Ataxia, could not be evaluated.”

#2 Was there a correlation between MRI findings suggestive of MSA-C and the SARA ataxia scale?

Reply: We agree with the reviewer’s point that assessing the correlation between MRI findings and cerebellar symptom rating scales, including SARA, is important. Unfortunately, no systematic assessment of cerebellar symptoms, including SARA, has been performed in the patients with MSA included in this study. We have added a note to the Discussion section about this as a limitation, as follows. We hope to evaluate the correlation between SARA and MRI findings in future prospective studies.

Page 21, line 348-351

“Third, owing to the retrospective nature of this study, the relationship between the MCP sT1W/T2W ratio and rapid eye movement sleep behavior disorder or clinical scales for cerebellar ataxia, such as the International Cooperative Ataxia Rating Scale or Scale for the Assessment and Rating of Ataxia, could not be evaluated.”

#3 Without a doubt, a prospective study would be very suitable to analyze the results of this study in greater depth, as well as with a greater number of cases of MSA-C and the group of patients with SCAs 3 and 6.

Reply: We agree with the reviewer’s comment and have revised and added sentences in the Discussion section, as follows.

Page 20, line 336

“This study has several limitations.”

Page 21, line 352-355

“Fourth, the sample size of the SCA3 group was small (n = 8), and future prospective studies with larger sample sizes that evaluate clinical scales for cerebellar ataxia are needed to validate the findings of this study and clarify the relationship between clinical severity of cerebellar ataxia and the MCP sT1W/T2W ratio.”

 

Reply to Reviewer 2

We greatly appreciate your helpful comments and suggestions. Changes made in accordance with reviewers’ comments are indicated in red font in the revised manuscript.

#1 One of the major concerns of the paper is the data used. The results might be biased mostly by noise. Data used in this work is 2D and image resolutions are not the same for T1w and T2w images and data was collected with 1.5 T MRI which hinherently has a low signal-to-noise ratio.

Reply: We agree with the reviewer’s concerns about bias in the data due to the MRI sequences and magnetic field strength of the MRI scanner used in this study. We have added sentences in the Discussion section as a limitation on this point, as follows.

Page 20, line 336 to Page 21 line 346

“The first is that 2D T2W images were co-registered with 3D T1W images, which do not have the same image resolution, and a 1.5-T MRI scanner with an inherently low signal-to-noise ratio was used; therefore, the results of the MCP sT1W/T2W ratio may be biased by noise. However, the MCP sT1W/T2W ratio values in patients with MSA-C correlated well with the extent of MCP hyperintensities and MCP volume, suggesting that the MCP sT1W/T2W ratio values measured in this study well reflected the degeneration of the MCP in MSA-C. Moreover, the diagnostic accuracy of the MCP sT1W/T2W ratio in discriminating MSA-C from controls in this study was comparable to that observed in a study using a 3-T MRI scanner [16]. Further studies employing 3D T1WI and T2WI sequences using a 3-T MRI scanner, which might allow better quantification of the sT1W/T2W ratio, are needed to confirm the results of this study.”

#2 The authors haven’t clearly mentioned what they are measuring with T1w/T2w and how sensitive the method is to tissue microstructure especially the myelin. T1w/T2w method is considered to be sensitive to myelin but depends on how T1w and T2w images are acquired as well as the image processing pipeline. For better quantification of T1w/T2w, 3D T1w and 3D T2w SPACE sequences are recommended. A recent work by Uddin et al (NMR in Biomedicine 2018) reported that TSE based T1w/T2w ratio is not a good measure to quantify myelin.

Reply: We want to thank the reviewer for this insightful comment. According to this reviewer’s comment, we have added sentences in the Discussion section, as follows.

Page 19, line 294-308

“Decreased MCP sT1W/T2W ratio values in patients with MSA-C might reflect demyelination and gliosis in the MCP due to MSA-C-related degeneration. Pathologically, the loss of axons, myelin, and reactive gliosis was observed in the MCP of patients with MSA-C [28]. The sT1W/T2W ratio is sensitive to myelin and decreases in demyelinating lesions [29, 30]. In contrast, inconsistent correlations with histology [31], myelin water imaging [12, 32], simultaneous tissue relaxometry, and magnetization transfer saturation index [33] suggest that the sT1W/T2W ratio is sensitive to not only myelin content but also other microstructural factors. Moreover, gliosis is presumed to affect the sT1W/T2W ratio in a downward direction, mainly by prolonged T1 relaxation time [34]. Therefore, although the sT1W/T2W ratio might not be specific for myelin content, it might be more sensitive than evaluating MCP hyperintensities on T2WI to detect degenerative processes in the MCP, such as demyelination and gliosis associated with MSA-C. In this study, the MCP sT1W/T2W ratio had better diagnostic accuracy in discriminating MSA-C from SCA3 than MCP volume. This result is consistent with those of previous pathological studies, showing that gliosis and demyelination precede atrophy in MSA [35, 36].”

In line with this revision, we have added references.

28. Itoh K, Kasai T, Tsuji Y, Saito K, Mizuta I, Harada Y, et al. Definite familial multiple system atrophy with unknown genetics. Neuropathology. 2014;34:309-13.

29. Glasser MF, Goyal MS, Preuss TM, Raichle ME, Van Essen DC. Trends and properties of human cerebral cortex: correlations with cortical myelin content. NeuroImage. 2014;93:165-75.

30. Nakamura K, Chen JT, Ontaneda D, Fox RJ, Trapp BD. T1-/T2-weighted ratio differs in demyelinated cortex in multiple sclerosis. Ann Neurol. 2017;82:635-9.

31. Righart R, Biberacher V, Jonkman LE, Klaver R, Schmidt P, Buck D, et al. Cortical pathology in multiple sclerosis detected by the T1/T2-weighted ratio from routine magnetic resonance imaging. Ann Neurol 2017;82:519-29.

32. Uddin MN, Figley TD, Marrie RA, Figley CR; For the CCOMS Study Group. Can T1w/T2w ratio be used as a myelin-specific measure in subcortical structures? Comparisons between FSE-based T1w/T2w ratios, GRASE-based T1w/T2w ratios and multi-echo GRASE-based myelin water fractions. NMR Biomed. 2018;31:e3868.

33. Hagiwara A, Hori M, Kamagata K, Warntjes M, Matsuyoshi D, Nakazawa M, et al. Myelin measurement: comparison between simultaneous tissue relaxometry, magnetization transfer saturation index, and T1w/T2w ratio methods. Sci Rep. 2018;8:10554.

34. Barnes D, McDonald WI, Landon DN, Johnson G. The characterization of experimental gliosis by quantitative nuclear magnetic resonance imaging. Brain. 1988;111:83-94.

35. Jellinger KA, Seppi K, Wenning GK. Grading of neuropathology in multiple system atrophy: proposal for a novel scale. Mov Disord. 2005;20:29-36.

36. Kon T, Mori F, Tanji K, Miki Y, Wakabayashi K. An autopsy case of preclinical multiple system atrophy (MSA-C). Neuropathology. 2013;33:667-72.

#3 The authors mentioned about T2 hyperintensities in MCP. I was wondering if they have any scores for WM hyperintensities or disease severity scores that can be correlated with T1w/T2w.

Reply: We agree with the reviewer’s point that assessing the correlation between MCP sT1W/T2W ratio values and scores for MCP hyperintensities or disease severity scores, such as SARA, is important. We have assessed the MCP hyperintensities semiquantitatively and evaluated the correlation between the MCP sT1W/T2W ratio and MCP hyperintensities. In line with this revision, we have added sentences in the Methods and Results sections. Unfortunately, no systematic assessment of cerebellar symptoms, including SARA, has been performed in the patients with MSA included in this study. We have added a note to the Discussion section about this as a limitation.

Page 7, line 159-161

“A board-certified neuroradiologist (H. Y.), who was blinded to the clinical data, semiquantitatively evaluated the extent of MCP hyperintensities in the patients with MSA by applying the following scores: 0 = absent, 1 = mild, 2 = moderate, and 3 = severe.”

Page 9, line 208-210

“Spearman correlation analysis was performed to evaluate the relationship between the extent of MCP hyperintensities assessed semiquantitatively and the MCP sT1W/T2W ratio values in patients with MSA-C.”

Page 12, line 256-258

“In patients with MSA-C, a significant inverse correlation between the extent of MCP hyperintensities and MCP sT1W/T2W ratio was observed (r = −0.745; p < 0.001).”

Page 21, line 348-351

“Third, owing to the retrospective nature of this study, the relationship between the MCP sT1W/T2W ratio and rapid eye movement sleep behavior disorder or clinical scales for cerebellar ataxia, such as the International Cooperative Ataxia Rating Scale or Scale for the Assessment and Rating of Ataxia, could not be evaluated.”

#4 MRI acquisition: data were acquired with a 1.5 T scanner. Please add information about the RF coil (i.e, 32-channel head coil), number of averages, number of slices.

Reply: According to the reviewer’s comments, we have revised sentences in the Discussion section, as follows.

Page 5, line 114-120

“All MR images were acquired using a 1.5-T MRI system (Genesis Signa; GE Healthcare) with a 12-channel head coil. The MRI parameters for T1WI were as follows: 3D-IR-SPGR; sagittal plane; TR, 8 ms; TE, 3 ms; flip angle, 15°; FOV, 220 × 220 mm; matrix, 256 × 256; voxel size, 0.86 × 0.86 × 1.5 mm; number of slices, 248; and number of averages, 1. Meanwhile, those for T2WI were as follows: 2D-TSE; TR, 4000 ms; TE, 106 ms; FOV, 220 × 220 mm; matrix, 320 × 256; voxel size, 0.43 × 0.43 × 6 mm; interslice gap, 1.5 mm; number of slices 24; and number of averages 1.”

#5 MRI preprocessing: “3D T1w images were linearly co-registered with 2D axial T2w images using SPM12. Brain masks were created by skull-stripping the co-registered T1w images using the BrainExtraction Tool with FSL (version 5.0.11) [20] and binarizing it with FSLmaths.”I was wondering why did the authors use SPM for coregistration only. It's better to use FSL or SPM, sometime the header information is not properly copied from one platform to other. However, it's possible to copy the headers from spm to fsl properly.

Reply: We originally planned to run the same pipeline as described by Cooper G, et al. (Front Neurol 2019;10:334). However, when testing the pipeline with our available data, we found that FSL did not coregister 3D and 2D images together very accurately, whereas SPM12 was able to do this very successfully.

#6 T1w/T2w calculation: The authors mentioned that for T1w/T2w quantification they used previously published technique by Misaki et al MRM 2014, but it is not clear how did they perform bias correction which is important for better quantification of T1w/T2w. Did they use their script or just follow their approach (it looks there is no publicly available tool online by Misaki et al)? Depending on the bias correction technique, T1w/T2w maps may contain residual radio frequency transmit field (B1+) biases, which may be correlated with these variables of interest, leading to potentially spurious results.

Reply: We thank the reviewer for pointing out that bias correction is missing our pipeline. We are aware of the growing literature demonstrating the importance of B1+ correction for T1W/T2W at 3 T and 7 T, where B1+ is more inhomogeneous. However, this study used a 1.5-T scanner, where B1+ is more homogenous and the effect could be assumed to be smaller (see, for example, Figure 3 of Dieringer et al. 2014 PLOS ONE, https://journals.plos.org/plosone/article?id=10.1371/journal.pone.0091318.

We have added a statement indicating a limitation that bias correction is missing in our pipeline, as follows.

Page 21, line 355-359

“Finally, bias correction was missing in our pipeline of sT1W/T2W calculation. Bias correction is important for the sT1W/T2W ratio on high-magnetic-field MRI to avoid the effects of the inhomogeneity of the radiofrequency transmit (B1+) field. However, this study used a 1.5-T scanner, where B1+ is more homogenous and the effect could be assumed to be smaller [49].”

In line with this revision, we have added a reference.

49. Dieringer MA, Deimling M, Santoro D, Wuerfel J, Madai VI, Sobesky J, et al. Rapid parametric mapping of the longitudinal relaxation time T1 using two-dimensional variable flip angle magnetic resonance imaging at 1.5 Tesla, 3 Tesla, and 7 Tesla. PLoS One. 2014;9:e91318.

#7 To measure diagnostic accuracy by Youden ROC method, I suspect about the accuracy of the measurement with this small sample size.

Reply: According to the reviewer’s comment, we have changed the method for determining the cutoff point for ROC analysis from the Youden method to the point on the ROC curve closest to the upper left corner. Moreover, we have added the small sample size as a limitation. With these changes, we have revised sentences in the Methods section and added sentences in the Results and Discussion sections, as follows.

Page 8, line 186-192

“All statistical analyses, except for receiver operating characteristic (ROC) curve analyses, were performed using Statistical Package for the Social Sciences (version 25.0; SPSS Inc., IBM Corp., Chicago, IL, USA). The ROC curve analyses and Smirnov–Grubb’s test were performed using EZR (Saitama Medical Center, Jichi Medical University, Saitama, Japan), which is a graphical user interface for R (The R Foundation for Statistical Computing, Vienna, Austria, version 2.13.0) [26]. More precisely, EZR is a modified version of R commander (version 1.6-3) designed to add statistical functions frequently used in biostatistics.”

Page 9, line 197-202

“For evaluating differences in MCP sT1W/T2W ratio values and MCP volume among the four groups, a one-way analysis of covariance was used using age as the covariate. The ability of the MCP sT1W/T2W ratio values and MCP volume to differentiate between groups of patients was assessed using ROC curve analyses, and the optimal cutoff point was determined as the point on the ROC curve closest to the upper left corner, as used by Holmes [27].”

Page 11, line 221-223

“The AUC for differentiating MSA-C from SCA-3 was 0.895 (95% confidence interval [CI], 0.794–0.995), with a sensitivity of 0.844 and specificity of 1.0.”

Page 11, line 224-227

“The AUC for differentiating MSA-C from controls was 0.960 (95% CI, 0.908–1.000), with a sensitivity of 0.906 and specificity of 0.941. The AUC for differentiating SCA-3 from SCA-6 was 0.898 (95% CI, 0.775–1.000), with a sensitivity of 0.875 and specificity of 0.812 (Fig. 3).”

Page 20, line 336

“This study has several limitations.”

Page 21, line 352-355

“Fourth, the sample size of the SCA3 group was small (n = 8), and future prospective studies with larger sample sizes that evaluate clinical scales for cerebellar ataxia are needed to validate the findings of this study and clarify the relationship between clinical severity of cerebellar ataxia and the MCP sT1W/T2W ratio.”

In line with these revisions, we have revised Figure 3 and added references.

26. Kanda Y. Investigation of the freely available easy-to-use software ‘EZR’ for medical statistics. Bone Marrow Transplant 2013;48:452-8.

27. Holmes W. A short, psychiatric, case-finding measure for HIV seropositive outpatients. Medical Care 1998;36:237-43.

#8 Figure1: you haven’t mentioned what type of images (i.e., T2-weighted images) are in the figure. Please include the corresponding T1w/T2w maps with the signal intensity scale.

Reply: According to the reviewer’s comments, we have revised Figure 1 and added a sentence in the Figure Legends section, as follows.

Page 7, line 171 to Page 8, line 172

“Upper row: axial T2-weighted images. Lower row: standardized T1-weighted/T2-weighted ratio maps”

#9 Figure2: significant difference between MSA-C with SCA3 and SCA6 might be derived from the outliers of MSA-C data. I would use the outlier removal tool and compare the results.

Reply: We thank the reviewer for the comment. We have added Smirnov–Grubb’s test for evaluating the outliers. Moreover, we have revised sentences in the Methods section and added a sentence in the Results section, as follows.

Page 8, line 188-191

“The ROC curve analyses and Smirnov–Grubb’s test were performed using EZR (Saitama Medical Center, Jichi Medical University, Saitama, Japan), which is a graphical user interface for R (The R Foundation for Statistical Computing, Vienna, Austria, version 2.13.0) [26].”

Page 9, line 212-213

“The Smirnov–Grubb’s test was used for evaluating the outliers of the MCP sT1W/T2W ratio value in each subject group.”

Page 11, line 227-231

“Even after excluding one patient with MSA-C with an outlier MCP sT1W/T2W ratio value, the MCP sT1W/T2W ratio values in patients with MSA-C were significantly lower than those in patients with SCA3 and SCA6 and controls (0.03 ± 0.06 vs. 0.13 ± 0.02, 0.17 ± 0.03, and 0.17 ± 0.03, respectively; p < 0.001).”

#10 Discussion section should be more organized.

Reply: According to the reviewer’s suggestion, we have revised sentences in the Discussion section, as follows.

Page 18, line 286-288

“We showed that the MCP sT1W/T2W ratio is highly sensitive and specific for not only distinguishing MSA-C from SCA3, SCA6, or controls but also differentiating SCA3 from SCA6.”

Page 19, line 309-310

“The usefulness of the MCP sT1W/T2W ratio in discriminating MSA-C from SCA3 in this study might be due to differences in the degree of MCP degeneration in the two diseases.”

Page 20, line 324-326

“The pathological background for this may be the difference in MCP degeneration, which is observed to some extent in SCA3, as described above, but is rarely observed in SCA6.”

#11 It would be great to show the comparison of volumetric measure of the MCP indifferent disease conditions.

Reply: We thank the reviewer for this important suggestion. The volume of the MCP was measured and compared between the groups. Moreover, we evaluated the correlation between MCP volume and MCP sT1w/T2w ratio value and compared the diagnostic accuracy between MCP volume and MCP sT1w/T2w ratio. Along with this revision, we have added sentences in the Methods and Results sections, as follows.

Page 8, line 174-183

“MCP volume analysis

The MCP volume was measured in all subjects by exploiting a validated probabilistic 3D atlas of the cerebellar white matter structure [24]. Parcellation at a 90% probability threshold was used. The atlas of the MCP was transformed from the Montreal Neurological Institute 152 space [20] to the individual subject’s space using the deformation parameters obtained from the normalization procedure using ANTs [23]. The mean volume value in the left and right MCPs was used as the MCP volume for each subject. For further analyses, the MCP volume was normalized by each subject’s intracranial volume. Values represent the ratio of the volume of the MCP divided by the subject-specific total intracranial volume (written as m × 10-3).”

Page 9, line 197-202

“For evaluating differences in MCP sT1W/T2W ratio values and MCP volume among the four groups, a one-way analysis of covariance was used using age as the covariate. The ability of the MCP sT1W/T2W ratio values and MCP volume to differentiate between groups of patients was assessed using ROC curve analyses, and the optimal cutoff point was determined as the point on the ROC curve closest to the upper left corner, as used by Holmes [27].”

Page 9, line 202-204

“The DeLong test was used to compare the area under curves (AUCs) between the MCP sT1W/T2W ratio and MCP volume to differentiate among the four groups.”

Page 9, line 210-212

“Pearson correlation analysis was performed to evaluate the relationship between the MCP sT1W/T2W ratio values and MCP volume in patients with MSA-C.”

Page 18, line 265-283

“The MCP volume, normalized by each subject’s intracranial volume, was significantly lower in patients with MSA-C patients than those in patients with SCA6 and controls (0.20 ± 0.06 vs. 0.31 ± 0.05 and 0.34 ± 0.05, respectively; p < 0.001). Moreover, the MCP volume was significantly lower in patients with SCA3 than those in patients with SCA6 and controls (0.23 ± 0.04 vs. 0.31 ± 0.05 and 0.34 ± 0.05, respectively; p < 0.001; p = 0.01 and p < 0.001).

The AUC value for differentiating MSA-C from SCA3 was 0.664 (95% CI, 0.486–0.842), with a sensitivity of 0.750 and specificity of 0.625. The AUC for differentiating MSA-C from SCA6 was 0.924 (95% CI, 0.852–0.995), with a sensitivity of 0.844 and specificity of 0.875. The AUC for differentiating MSA-C from controls was 0.967 (95% CI, 0.925–1.000), with a sensitivity of 0.906 and specificity of 0.941). The AUC for differentiating SCA3 from SCA6 was 0.914 (95% CI, 0.802–1.000), with a sensitivity of 0.875 and specificity of 0.875. The AUC of the MCP sT1W/T2W ratio was greater than that of the MCP volume for differentiating MSA-C from SCA3 (p = 0.0049). No significant differences were observed between the AUC of the MCP sT1W/T2W ratio and that of MCP volume for differentiating MSA-C from SCA6 and controls (p = 0.304 and 0.781, respectively). Moreover, no significant difference was observed between the AUC of the MCP sT1W/T2W ratio and that of MCP volume for differentiating SCA3 from SCA6 (p = 0.855). In patients with MSA-C, a significant correlation was observed between the MCP volume and MCP sT1W/T2W ratio (r = 0.732; p < 0.001).”

#12 Please add the small sample size as a limitation.

Reply: We agree with the reviewer’s concerns about the small sample sizes in our study. We have revised and added sentences in the Discussion section, as follows.

Page 20, line 336

“This study has several limitations.”

Page 21, line 352-355

“Fourth, the sample size of the SCA3 group was small (n = 8), and future prospective studies with larger sample sizes that evaluate clinical scales for cerebellar ataxia are needed to validate the findings of this study and clarify the relationship between clinical severity of cerebellar ataxia and the MCP sT1W/T2W ratio.”

#13 “In conclusion, the MCP sT1w/T2w ratio is a highly sensitive imaging-based marker for detecting MSA-C related changes and might be useful for differentiating 274 MSA-C from SCA3 or SCA6.” With a relatively small sample size and poor quality of data (especially 2D T1w,T2w data, 1.5 mm vs. 6mm slice thickness collected at 1.5 T), it is tough to claim “highly sensitive imaging-based marker ….”

Reply: We thank the reviewer for the comment. Accordingly, we have revised sentences in the Abstract and Conclusions sections, as follows.

Page 3, line 50-51

“The MCP sT1W/T2W ratio might be a sensitive imaging-based marker for detecting MSA-C-related changes and differentiating MSA-C from SCA3 or SCA6.”

Page 21, line 362 to Page 22, line 363

“In conclusion, the MCP sT1W/T2W ratio might be a sensitive imaging-based marker for detecting MSA-C-related changes and useful for differentiating MSA-C from SCA3 or SCA6.”

---

## [Decision Letter · Decision Letter 1]

18 Nov 2021

PONE-D-21-24426R1Diagnostic efficacy of the magnetic resonance T1w/T2w ratio for the middle cerebellar peduncle in multiple system atrophy and spinocerebellar ataxiaPLOS ONE

Dear Dr. Sugiyama,

Thank you for submitting your manuscript to PLOS ONE. After careful consideration, we feel that it has merit but does not fully meet PLOS ONE’s publication criteria as it currently stands. Therefore, we invite you to submit a revised version of the manuscript that addresses the points raised during the review process.

While one of the Reviewers feels that your manuscript is ready to be accepted, the other Reviewer still has major concerns that I invite you to address. While the sample size limitation can be acknowledged in the manuscript, I strongly recommend that you follow the Reviewer's advice to implement bias field correction in the analysis.

We look forward to receiving your revised manuscript.

Kind regards,

Niels Bergsland

Academic Editor

PLOS ONE

Reviewers' comments:

Reviewer's Responses to Questions

**Comments to the Author**

1. If the authors have adequately addressed your comments raised in a previous round of review and you feel that this manuscript is now acceptable for publication, you may indicate that here to bypass the “Comments to the Author” section, enter your conflict of interest statement in the “Confidential to Editor” section, and submit your "Accept" recommendation.

Reviewer #1: All comments have been addressed

Reviewer #2: (No Response)

2. Is the manuscript technically sound, and do the data support the conclusions?

Reviewer #1: Yes

Reviewer #2: Partly

3. Has the statistical analysis been performed appropriately and rigorously? 

Reviewer #1: Yes

Reviewer #2: Yes

4. Have the authors made all data underlying the findings in their manuscript fully available?

Reviewer #1: Yes

Reviewer #2: Yes

5. Is the manuscript presented in an intelligible fashion and written in standard English?

Reviewer #1: Yes

Reviewer #2: Yes

6. Review Comments to the Author

Reviewer #1: The revised manuscript is now OK for publication. I have no additional comments or suggestions fo rthe authors !

Reviewer #2: The authors have partly responded to my questions and made changes to the manuscript. I would thank the authors to add healthy controls. But the major concern is the data used and sample size. Therefore, it can be reported as preliminary findings and further investigation with a large sample size and standard T1 and T2 images is required to validate these findings. I have some comments:

1. The authors followed the paper by Misaki et al 2015 but they didn’t perform bias correction. As I mentioned before that bias correction is an important step for T1w/T2w ratio mapping and depending on the bias correction technique, T1w/T2w maps may contain residual radiofrequency B1+ biases, which may be correlated with these variables of interest, leading to potentially spurious results. It's true that B1+ is more homogeneous at 1.5T compared to higher magnetic field strengths such as 3T or 7T (say center brightness artifact is ~30% at 3T while ~1.5% at 1.5T; Bernstein et al 2006), but not perfectly homogeneous and it’s still required to perform bias correction for T1w/T2w mapping even at 1.5 T MRI (see previous works at 1.5 T; Ganzetti et al 2016).

2. It would be great to add a figure for a pipeline for creating a T1w/T2w image (i.e., summarize key steps with representative images in native and standard spaces).

3. Figure 1: Why did the authors use images at different slice locations in Figure 1? I would recommend using the same slice locations for all cases in Figure 1. Also, show the corresponding images for T1w/T2w maps with the same image size.

4. Abstract: “Methods” section was not updated for healthy controls.

7. PLOS authors have the option to publish the peer review history of their article (what does this mean?). If published, this will include your full peer review and any attached files.

Reviewer #1: No

Reviewer #2: **Yes: **Md Nasir Uddin

---

## [Author Response · Author response to Decision Letter 1]

24 Mar 2022

Reply to Reviewer 1

We greatly appreciate your helpful comments and suggestions. Changes made in accordance with reviewers’ comments are indicated in red font in the revised manuscript.

#1 The authors have partly responded to my questions and made changes to the manuscript. I would thank the authors to add healthy controls. But the major concern is the data used and sample size. Therefore, it can be reported as preliminary findings and further investigation with a large sample size and standard T1 and T2 images is required to validate these findings. I have some comments:

Reply: We thank the Reviewer for their important comment. Accordingly, we have revised the title.

Page1, line 1-3

“Diagnostic efficacy of the magnetic resonance T1w/T2w ratio for the middle cerebellar peduncle in multiple system atrophy and spinocerebellar ataxia: a preliminary study”

#2 The authors followed the paper by Misaki et al 2015 but they didn’t perform bias correction. As I mentioned before that bias correction is an important step for T1w/T2w ratio mapping and depending on the bias correction technique, T1w/T2w maps may contain residual radiofrequency B1+ biases, which may be correlated with these variables of interest, leading to potentially spurious results. It's true that B1+ is more homogeneous at 1.5T compared to higher magnetic field strengths such as 3T or 7T (say center brightness artifact is ~30% at 3T while ~1.5% at 1.5T; Bernstein et al 2006), but not perfectly homogeneous and it’s still required to perform bias correction for T1w/T2w mapping even at 1.5 T MRI (see previous works at 1.5 T; Ganzetti et al 2016).

Reply: We followed the reviewer's suggestion and added the bias collection step, which partially changed the results of the study. We strongly agree with the reviewer's comments on the importance of bias collection step and would like to express our deepest gratitude to the reviewer for suggesting the addition of bias collection. Accordingly, we have revised and added sentences in the Abstract, Methods, Results, and Discussion sections.

Page 2, line 41-48

“The MCP sT1w/T2w ratio showed high diagnostic accuracy for distinguishing MSA-C from SCA3 (area under curve = 0.934), SCA6 (area under curve = 0.965), and controls (area under curve = 0.980). The diagnostic accuracy of the MCP sT1w/T2w ratio for differentiating MSA-C from SCA3 or SCA6 (90.0% for MSA-C vs. SCA3, and 91.7% for MSA-C vs. SCA6) was comparable to or superior than that of visual interpretation of MCP hyperintensities (80.0–87.5% in MSA-C vs. SCA3 and 87.6–97.9% in MSA-C vs. SCA6) or the HCB sign (72.5–80.0% in MSA-C vs. SCA3 and 77.1–93.8% in MSA-C vs. SCA6).”

Page 6, line 123-125

“All T1w and T2w images were preprocessed as previously described before the sT1w/T2w ratio was calculated [16]; specifically, the intensity inhomogeneity correction was applied to the T1w and T2w images using N4BiasFieldCorrection [20].”

Page 12, line 226-238

“The MCP sT1w/T2w ratio values were significantly lower in patients with MSA-C than those in patients with SCA3 and SCA6 and controls (0.07 ± 0.06 vs. 0.17 ± 0.03, 0.18 ± 0.03, and 0.19 ± 0.03, respectively; p < 0.001) (Fig. 3). The AUC for differentiating MSA-C from SCA-3 was 0.934 (95% confidence interval [CI], 0.854–1.000), with a sensitivity of 0.906 and specificity of 0.875. The AUC value for differentiating between MSA-C and SCA-6 was 0.965 (95% CI, 0.922–1.000), with a sensitivity of 0.906 and a specificity of 0.938. The AUC for differentiating MSA-C from controls was 0.980 (95% CI, 0.951–1.000), with a sensitivity of 0.906 and specificity of 0.941. The AUC for differentiating SCA-3 from SCA-6 was 0.633 (95% CI, 0.388–0.878), with a sensitivity of 0.750 and specificity of 0.625 (Fig. 4). Even after excluding one patient with MSA-C with an outlier MCP sT1w/T2w ratio value, the MCP sT1W/T2W ratio values in patients with MSA-C were significantly lower than those in patients with SCA3 and SCA6 and controls (0.07 ± 0.05 vs. 0.17 ± 0.03, 0.18 ± 0.03, and 0.19 ± 0.03, respectively; p < 0.001).”

Page 13, line 263-265

“In patients with MSA-C, a significant inverse correlation between the extent of MCP hyperintensities and MCP sT1w/T2w ratio was observed (r = −0.832; p < 0.001).”

Page 16, line 267-274

“Diagnostic accuracy of the MCP sT1w/T2w ratio, visual interpretation of the MCP hyperintensities, and the HCB sign are listed in Table 3, and we found that the diagnostic performance of the MCP sT1w/T2w ratio for differentiating MSA-C from SCA3, SCA6, and controls was comparable to or better than that of visual interpretation of MCP hyperintensities and the HCB sign for all three reviewers. However, the diagnostic performance of visual interpretation of MCP hyperintensities and the HCB sign for differentiating SCA3 from SCA6 was comparable to or better than that of the MCP sT1w/T2w ratio.”

Page 19, line 286-292

“The AUC of the MCP sT1w/T2w ratio was greater than that of the MCP volume for differentiating MSA-C from SCA3 (p < 0.001). No significant differences were observed between the AUC of the MCP sT1w/T2w ratio and that of MCP volume for differentiating MSA-C from SCA6 and controls (p = 0.234 and 0.577, respectively). The AUC of the MCP volume was greater than that of the MCP sT1w/T2w ratio for differentiating SCA3 from SCA6 (p = 0.020).”

Page 19, line 292-293

“In patients with MSA-C, a significant correlation was observed between the MCP volume and MCP sT1W/T2W ratio (r = 0.619; p < 0.001).”

Page 19, line 296-297

“We showed that the MCP sT1W/T2W ratio is highly sensitive and specific for distinguishing MSA-C from SCA3, SCA6, or controls.”

Page 21, line 332-336

“The substantial discriminatory power of the MCP sT1w/T2w ratio and visual interpretation of MCP hyperintensities and the HCB sign described here implies that they may also be useful for differentiating between SCA3 and SCA6. The pathological background for this may be the difference in MCP degeneration, which is observed to some extent in SCA3, as described above, but is rarely observed in SCA6.”

Page 21, line 341-345

“The cohort of SCA3 and SCA6 patients included in this study is not large, and therefore, further studies that include data from a large cohort of SCA3 and SCA6 patients are needed to verify the usefulness of MCP sT1w/T2w ratio values and visual interpretation of MCP hyperintensities and the HCB sign in differentiating between SCA3 and SCA6.”

Page 22, line 362-365

“Finally, the sample size of the SCA3 group was small (n = 8), and future prospective studies with larger sample sizes that evaluate clinical scales for cerebellar ataxia are needed to validate the findings of this study and clarify the relationship between clinical severity of cerebellar ataxia and the MCP sT1W/T2W ratio.”

According to this revision, we have revised Figure 3 and Figure 4.

According to this revision, we have revised S1 Table.

According to this revision, we have added a reference.

20. Tustison NJ, Avants BB, Cook PA, Zheng Y, Egan A, Yushkevich PA, et al. N4ITK: improved N3 bias correction. IEEE Trans Med Imaging. 2010;29:1310-20.

#3 It would be great to add a figure for a pipeline for creating a T1w/T2w image (i.e., summarize key steps with representative images in native and standard spaces).

Reply: We thank the Reviewer for their important comment. Accordingly, we have added a pipeline for creating a sT1w/T2w ratio map as Figure 1.

Page 6, line 140-141

“Schematic representation of the pipeline for creating a sT1w/T2w ratio map is described in Figure 1.”

Page 7, line 152-154

“Figure 1. Schematic representation of the pipeline for creating a sT1w/T2w ratio map.

T1w, T1-weighted; T2w, T2-weighted; GM, gray matter: WM, white matter; sT2, a scaled T2-weighted image”

#4 Figure 1: Why did the authors use images at different slice locations in Figure 1? I would recommend using the same slice locations for all cases in Figure 1. Also, show the corresponding images for T1w/T2w maps with the same image size.

Reply: We thank the Reviewer for their important comment. Accordingly, we have revised Figure 2 and a sentence in Figure legend.

Page 8, line 173-175

“(A) Grade 0 (negative) HCB sign and absent MCP hyperintensities in a 74-year-old man with a SCA6 disease duration of 2.3 years.”

#5 Abstract: “Methods” section was not updated for healthy controls.

Reply: We thank the Reviewer for their careful observation and apologize for the oversight. The sentence has been revised as follows:

Page 2, line 34-35

“We included 32 MSA-C, 8 SCA type 3 (SCA3), 16 SCA type 6 (SCA6) patients, and 17 controls, and the MCP sT1w/T2w ratio was analyzed using a region-of-interest approach.”

---

## [Decision Letter · Decision Letter 2]

1 Apr 2022

Diagnostic efficacy of the magnetic resonance T1w/T2w ratio for the middle cerebellar peduncle in multiple system atrophy and spinocerebellar ataxia: a preliminary study

PONE-D-21-24426R2

Dear Dr. Sugiyama,

We’re pleased to inform you that your manuscript has been judged scientifically suitable for publication and will be formally accepted for publication once it meets all outstanding technical requirements.

Kind regards,

Niels Bergsland

Academic Editor

PLOS ONE

Additional Editor Comments (optional):

Reviewers' comments:

Reviewer's Responses to Questions

**Comments to the Author**

1. If the authors have adequately addressed your comments raised in a previous round of review and you feel that this manuscript is now acceptable for publication, you may indicate that here to bypass the “Comments to the Author” section, enter your conflict of interest statement in the “Confidential to Editor” section, and submit your "Accept" recommendation.

Reviewer #2: All comments have been addressed

2. Is the manuscript technically sound, and do the data support the conclusions?

Reviewer #2: Yes

3. Has the statistical analysis been performed appropriately and rigorously? 

Reviewer #2: Yes

4. Have the authors made all data underlying the findings in their manuscript fully available?

Reviewer #2: (No Response)

5. Is the manuscript presented in an intelligible fashion and written in standard English?

Reviewer #2: (No Response)

6. Review Comments to the Author

Reviewer #2: The authors have sufficiently improved the manuscript, and I think the manuscript is now ready for publication.

7. PLOS authors have the option to publish the peer review history of their article (what does this mean?). If published, this will include your full peer review and any attached files.

Reviewer #2: **Yes: **Md Nasir Uddin

---

## [Editor Report · Acceptance letter]

7 Apr 2022

PONE-D-21-24426R2 

Diagnostic efficacy of the magnetic resonance T1w/T2w ratio for the middle cerebellar peduncle in multiple system atrophy and spinocerebellar ataxia: a preliminary study 

Dear Dr. Sugiyama:

I'm pleased to inform you that your manuscript has been deemed suitable for publication in PLOS ONE. Congratulations! Your manuscript is now with our production department. 

Kind regards, 

on behalf of

Dr. Niels Bergsland 

Academic Editor

PLOS ONE